biomedical engineering/biomaterials

atomic force microscopy, coronary arteries, endothelium, multiscale, scanning electron microscopy, surface roughness

**Author for correspondence:**
Hanna E. Burton
e-mail: h.e.burton@bham.ac.uk

# Multiscale three-dimensional surface reconstruction and surface roughness of porcine left anterior descending coronary arteries

Hanna E. Burton[1,2], Rachael Cullinan[3], Kyle Jiang[4] and Daniel M. Espino[2]

[1]PDR – International Centre for Design and Research, Cardiff Metropolitan University, Cardiff CF5 2YB, UK
[2]Biomedical Engineering Research Group, Department of Mechanical Engineering,
[3]School of Chemical Engineering, and [4]Research Centre for Micro/Nanotechnology, Department of Mechanical Engineering, University of Birmingham, Birmingham B15 2TT, UK

HEB, 0000-0002-7763-703X; RC, 0000-0003-4503-9286

The aim of this study was to investigate the multiscale surface roughness characteristics of coronary arteries, to aid in the development of novel biomaterials and bioinspired medical devices. Porcine left anterior descending coronary arteries were dissected *ex vivo*, and specimens were chemically fixed and dehydrated for testing. Surface roughness was calculated from three-dimensional reconstructed surface images obtained by optical, scanning electron and atomic force microscopy, ranging in magnification from 10× to 5500×. Circumferential surface roughness decreased with magnification, and microscopy type was found to influence surface roughness values. Longitudinal surface roughness was not affected by magnification or microscopy types within the parameters of this study. This study found that coronary arteries exhibit multiscale characteristics. It also highlights the importance of ensuring consistent microscopy parameters to provide comparable surface roughness values.

# 1. Introduction

Cardiovascular diseases are the leading cause of mortality worldwide [1]. Future replacement materials to treat cardiovascular diseases would benefit from being biomimetic [2]. To aid in the development of bioinspired materials, it is important to assess

biological structures at multiple scales to allow replacement materials to replicate the native tissue at both a micro- and nano-scale.

Recently, the arithmetic average of surface roughness ($Ra$) has been used to enable the quantification of blood vessel surface [3,4] in the circumferential and longitudinal direction; previous to this, all analysis had been qualitative [5,6]. Studies by the authors have discussed the importance of a correction factor when measuring surface roughness of processed tissue [4]. The surface roughness of other biological tissue, such as articular cartilage [7,8], has been investigated without the use of a correction factor. These studies have noted a multiscale variance when assessing the surface roughness of biological surfaces [8]. It is unknown whether a similar relationship is noted for surface roughness of coronary arteries.

The arithmetic mean surface roughness has been measured by light microscopy [3]. However, the maximum magnification of the light microscope system used was 100×. For a higher magnification, an alternate system such as scanning electron microscopy (SEM) or atomic force microscopy (AFM) is preferred, as these techniques can capture the surface at a micro- and nano- scale. AFM has been used to assess atherosclerosis on artery walls [5] but provided no quantitative measurement of the surface. However, different microscopy techniques can provide dissimilar results [7]. For a comparison of magnification across a variety of microscopy techniques, a like-for-like magnification should ideally be established to provide an overlap between results.

Disease of biological tissue can result in changes to surface roughness [9,10], and the potential of using multiscale biological and physiological properties has been demonstrated in the heart for creating computational models of healthy and diseased circulatory systems [11]. Understanding the multiscale variance in the surface roughness of coronary arteries would provide a quantifiable boundary condition to enable multiscale computational fluid dynamic (CFD) modelling, which could assist in predicting helical blood flow and disease [12]. Further, physical models have been created through additive manufacturing to study blood flow in healthy and diseased cardiovascular systems [13] and to mimic the mechanical properties of arteries to predict leakage after valve replacement [14]. This demonstrates the potential for creating bioinspired replacement materials through computer aided manufacture, and also phantoms which could mimic the multiscale surface of coronary arteries to study the effect of disease on blood flow.

The aim of this study is to assess the multiscale variation of surface roughness of left anterior descending (LAD) coronary arteries, in both the circumferential and longitudinal orientation. The methods of optical microscopy, SEM and AFM are considered to allow multiscale comparison of surface roughness between 10× to 2000× magnification. As measurement technique is known to alter the measured surface roughness [8], this study includes overlap of magnification with microscopy type. Results are presented for individual microscopy techniques, and trends of surface roughness are investigated between microscopy techniques and magnification. The correction factor for chemical tissue processing calculated in previous work [4] was applied to results to provide outer limits of circumferential surface roughness.

# 2. Methods

## 2.1. Sample preparation

Eight porcine hearts ($N = 8$) were supplied by Fresh Tissue Supplies (Horsham, UK). Hearts were frozen on excision. No animals were specifically sacrificed for this study. Hearts were defrosted at approximately 4°C overnight before dissection. The LAD coronary artery was identified and dissected (figure 1a). A longitudinal incision (along the length of the artery) was made along the LAD sample to expose the internal surface. Excess cardiac muscle tissue was removed from samples leaving coronary artery tissue only. Finally, the sample was sectioned into three specimens of 20 mm each (figure 1b), categorized as proximal, middle and distal, where in this case proximal refers to a position nearer the base of the heart and distal near to the apex of the heart, along a longitudinal axis the LAD. The dissection process is described in previous work [3]. For this study, the middle specimens were selected for investigation (these are identified in table 1), as there is no change in surface roughness along the length of the LAD coronary artery [3], enabling surface roughness to be assessed in the circumferential and longitudinal directions (figure 1). When referring to the number of hearts, the notation $N$ is used. When referring to number of specimens, $n$ is used.

To allow a comparison between magnification and microscopic technique, tissue samples underwent fixation and dehydration, following a standard protocol for soft mammalian tissues [15]. This process is described in more detail elsewhere [4]; briefly, specimens were immersed in a 3% glutaraldehyde solution (Fluka Analytical, Sigma Aldrich, St Louis, MO, USA) with 0.2 M sodium phosphate buffer for 1 h.

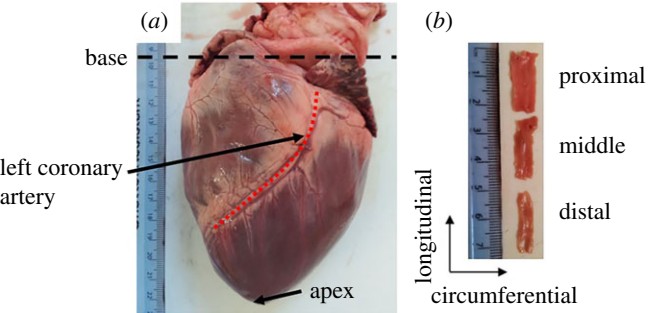

**Figure 1.** (*a*) Defrosted heart pre-dissection, with apex, base and left coronary artery identified. (*b*) LAD specimens prepared as 20 mm sections, with longitudinal and circumferential axes labelled.

**Table 1.** Specimen selection. In heart identification, the letters a–h identify which specimens were used for which microscopy techniques. Profile roughness parameter, *Ra*, is the arithmetic mean deviation of roughness profile.

| microscopy type | heart identification | scan area | number of repeat *Ra* (in each orientation) | magnifications |
|---|---|---|---|---|
| optical | $N = 6$ (a, b, c, d, e, f) | 1623 × 1623 µm | | 10 |
| | | 811 × 811 µm | 5 | 20 |
| | | 323 × 323 µm | | 50 |
| | | 162 × 162 µm | | 100 |
| scanning electron | $N = 5$ (d, e, f, g, h) | 1230 × 1700 µm | | 100 |
| | | 123 × 170 µm | 5 | 1000 |
| | | 61 × 85 µm | | 2000 |
| atomic force | $N = 3$ (a, b, c) | 50 × 32 µm | 3 | 5500[a] |

[a]AFM magnification is not a true magnification, but a calculated equivalent magnification.

Subsequently, specimens were rinsed in three 10 min washes of phosphate buffer saline (PBS) solution to remove any remaining glutaraldehyde. To ensure that the samples remained hydrated they were stored in PBS solution at 4°C until dehydration.

Dehydration was performed in washes of 10 min with increasing concentrations of ethanol (Fisher Chemical, Fisher Scientific UK Ltd, Loughborough, UK) at 30%, 50%, 70%, 95% and two washes at 100%. Finally, hexamethyldisilasane (HMDS; Aldrich Chemistry, St Louis, MO, USA) was used to complete dehydration, removing any remaining ethanol from the specimen by displacement. The specimen underwent a wash of HMDS for 15 min before replenishing with fresh HMDS to be left overnight to evaporate.

## 2.2. Outline of microscopy

To allow a multiscale analysis of surface roughness, magnification was varied from 10× to 2000× across microscopy techniques (light microscopy and SEM), and included an equivalent magnification of 5500× using AFM (table 1). Six specimens were assessed by optical imaging, of which five of these specimens were investigated by SEM (table 1). The remaining specimen, and two further specimens were assessed by AFM (table 1). Due to charging damage associated with SEM, it was not possible to re-use SEM samples for AFM. The maximum magnification lens with optical microscopy was 100×, therefore, the minimum magnification investigated via optical microscopy was chosen at an order of magnitude lower (10× magnification). This builds on previous work by the authors [3], mostly focused on 10× magnification.

To assess an overlap of scale, SEM was investigated with a minimum magnification of 100× to allow like-for-like comparison with optical microscopy. Again, to assess multiscale effects of surface roughness, surface imaging was performed at an order of magnitude greater than this (1000×), and at a magnification of 2000×. AFM was used to gain a greater equivalent magnification (5500×).

## 2.3. Surface roughness

Previously, the surface roughness of coronary arteries has been quantified using the arithmetic mean deviation of roughness profile, $Ra$ [3,4]. Further information on calculating $Ra$ is presented in earlier work [3]. In this study, $Ra$ was measured along profiles drawn across the entire length of the reconstructed images in both the longitudinal ($Ra_L$) and circumferential direction ($Ra_C$) of the artery, for both for optical and SEM (table 1). The mean of five measurements was taken in each direction for each specimen. For AFM, the same $Ra$ profiles were taken but the mean of three values was calculated. This used the same averaging process of assessing $Ra$ in three different regions, as presented in Tholt *et al.*'s study [16]. When measuring $Ra$, certain exclusion criteria were set for drawing profiles. Bifurcation 'holes', where smaller vessels connected to the artery, were avoided, as they formed part of the blood vessel structure, not surface topology. Further, edges of specimens damaged due to dissection, and areas of minor residue left by the processing of tissue, were avoided as they are not intrinsic properties of the surface of the artery.

## 2.4. Optical imaging

Using an optical focus variation microscope (G5 Infinite Focus, Alicona UK, Kent, UK), images were taken at 10×, 20×, 50× and 100× magnifications ($n = 6$). The Alicona Infinite Focus microscope is a non-contact, optical, three-dimensional (3D) measurement system. Scanning was performed between the maximum and minimum focusing positions of the height of each sample in the $z$ plane through focusing of the lens. Similarly, the area of the scan was controlled by selecting the maximum and minimum $x$ and $y$ positions of the sample. Note, the $x$ and $y$ axes are parallel to the circumferential and longitudinal directions, respectively, and the $z$ axis is perpendicular to the $x$-$y$ plane (i.e. aligned parallel to the direction of the thickness of the sample). Lighting was controlled via white light emitting diode coaxial illumination. Illumination intensity, and lateral ($x$ and $y$ axis) and vertical ($z$ axis) resolution were adjusted using automated idealized settings, consistent with previous studies [4,17].

The area of the scan (ranging from 0.16 to 1.62 µm$^2$) was selected as the automatic region at each magnification to reduce scan time and file size (table 1). Areas were selected to avoid bifurcations and damage due to dissection when selecting the 10× magnification zone. Scan areas were taken at the same central $x$ and $y$ positions as the 10× magnification area, simply increasing the magnification at the centre of each image. 3D reconstruction was performed using the Alicona IF-Laboratory Measurement Module (version 6.1, Alicona UK, Kent, UK), from which $Ra$ was measured.

## 2.5. Scanning electron microscopy

Specimens were imaged using a Hitachi TM3030 SEM (Hitachi Ltd., Tokyo, Japan) at 15 kV. Surfaces were scanned at 100× magnification, at 1000× magnification and 2000× magnification ($n = 5$). Under vacuum conditions, an electron beam was focused on the surface of the sample. In addition to preparing specimens through fixation and dehydration described previously [4], for SEM, specimens underwent sputter coating. Prior to imaging samples using SEM, specimens were mounted on an SEM stub with conductive double-sided tape and were sputter coated with gold at 2.5 kV using an Agar automatic sputter coater (Agar Scientific, Elektron, Technology UK Ltd, Essex, UK) for 30 s at 20 mA, followed by a further 30 s at 30 mA, to ensure an even covering of 150–200 Å [15].

For 3D surface reconstruction, four two-dimensional (2D) surface scans are stitched together using 3D-VIEW software (Hitachi Ltd., Tokyo, Japan) [18]. $Ra$ was measured from the 3D reconstructed surfaces.

## 2.6. Atomic force microscopy

Specimens (table 1) were measured using a Nanowizard II AFM (JPK Instruments AG, Berlin, Germany), operating in non-contact tapping mode, using height profile modulation. PointProbe Plus (PPP-NCR) non-contact high resonance frequency silicon scanning mode microscopy sensors were employed, with tip radius less than 10 nm (Nanosensors TM., Switzerland), similar to the 8 nm tip radius used by Timashev *et al.* when studying arteries with AFM [5]. The tapping mode was chosen due to tissue deformation associated with AFM in contact mode [19], and to avoid a stick-slip phenomenon [20]. Further, using tapping mode allowed for a more accurate representation of the topographical area when processed into 3D surface profiles.

The tip height was 10–15 µm, with a cantilever force constant of 10–130 N m$^{-1}$. An area of 50 µm × 32 µm was scanned, at an equivalent magnification of 5500×. Scans were processed using JPK Data Processing software (JPK Instruments AG, Berlin, Germany). The mean of three values was calculated for $Ra$ in both the circumferential and longitudinal direction for each sample.

## 2.7. Correction factor

The correction factor presented previously [4] was calculated for specimens where a significant difference was seen in $Ra$ before and after processing. Its purpose was to present the outer limits of $Ra$ values. Thus, the correction factor (equation (2.1)) was applied to surface roughness in the circumferential direction to give $Ra_{C\beta}$, as a significant difference was seen in $Ra_C$ due to processing [3,4].

$$Ra_\beta = \frac{Ra}{1.46}. \tag{2.1}$$

## 2.8. Data analysis

Two separate analyses were performed on data: the first assessed if magnification caused a significant difference to surface roughness; the second assessed if microscopic technique caused a significant difference to surface roughness. For this, the null hypothesis of a one-way analysis of variance (ANOVA) was assessed, with Tukey pairwise post hoc test used to assess a significant difference ($p < 0.05$) in surface roughness due to magnification with both optical microscopy and SEM separately. Additionally, significant differences in surface roughness were assessed ($p < 0.05$) in magnification groups across microscopic techniques combined, which also included results measured by AFM [21,22].

Finally, surface roughness in the circumferential and longitudinal direction was assessed for significant difference ($p < 0.05$), testing the null hypothesis using a student paired $t$-test at all magnifications with each microscopy technique. Data is presented as mean ± standard deviation unless otherwise stated. The correlation of surface roughness was assessed, with significance assessed for a relationship fit ($p < 0.05$).

# 3. Results

## 3.1. Optical imaging results

Using magnifications between 10 × to 100× with the optical microscope method did not significantly alter the measured $Ra$ value ($p > 0.05$; table 2). When measuring surface roughness from 3D reconstructed optical images, $Ra_C$ was significantly greater than $Ra_L$ at all magnification levels (0.72 ± 0.31 µm and 0.28 ± 0.10 µm; table 2; $p < 0.05$). 2D images at each optical magnification, and corresponding 3D reconstruction, are shown in figure 2 and figure 3, respectively. From these figures, ridges are visible along the longitudinal direction.

## 3.2. Scanning electron microscopy results

Using magnifications between 100× to 2000× with SEM did not affect $Ra$ values ($p > 0.05$; table 2). However, unlike optical microscopy, when using SEM to measure surface roughness there was no significant difference between $Ra_C$ and $Ra_L$ (0.21 ± 0.09 µm and 0.19 ± 0.08 µm; table 2; $p > 0.05$) for each magnification. At 100× magnification, there was no significant difference in $Ra$ (both longitudinally and circumferentially) between optical microscopy and SEM results ($p > 0.05$; table 2). 3D reconstructed and 2D SEM images are shown in figure 4 and figure 5, respectively, where ridges are visible in the longitudinal direction at all magnifications. Again, minor residue due to processing is present on the surface of specimens causing artefacts. Additionally, in the 2D reconstruction slight charging is apparent (figure 5), and cracks can be seen on the surface of the specimen due to processing for SEM.

## 3.3. Atomic force microscopy results

When using AFM to measure surface roughness there was no significant difference between $Ra_C$ and $Ra_L$ (0.50 ± 0.04 µm and 0.34 ± 0.02 µm; table 2; $p > 0.05$). Figures 6 and 7 show a 2D surface image and the corresponding 3D reconstruction, with the longitudinal direction (ridge alignment) identified in figure 6. Minor residue from the processing of tissue is apparent on the surface of the specimen

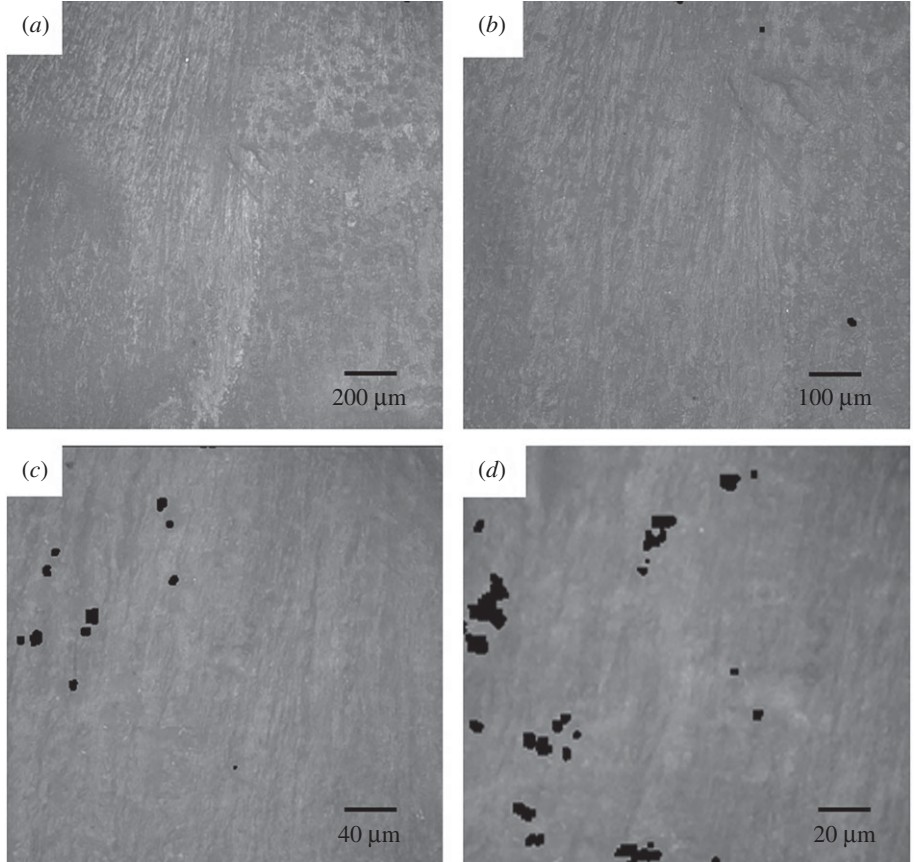

**Figure 2.** 2D optical images using Alicona Infinite Focus microscope at (*a*) 10×, (*b*) 20×, (*c*) 50× and (*d*) 100× magnification.

**Table 2.** Surface roughness values at each magnification and microscopy type. Results are shown as mean ± s.d. For: optical microscopy $n = 6$; SEM $n = 5$; and AFM $n = 3$.

|  | $Ra_C$ (μm) | $Ra_L$ (μm) | $Ra_{C\beta}$ (μm) |
|---|---|---|---|
| 10× optical | 0.91 ± 0.27 | 0.35 ± 0.08 | 0.62 ± 0.17 |
| 20× optical | 0.73 ± 0.32 | 0.25 ± 0.07 | 0.50 ± 0.20 |
| 50× optical | 0.69 ± 0.36 | 0.26 ± 0.09 | 0.47 ± 0.22 |
| 100× optical | 0.55 ± 0.27 | 0.26 ± 0.13 | 0.38 ± 0.17 |
| 100× SEM | 0.19 ± 0.08 | 0.19 ± 0.05 | 0.13 ± 0.05 |
| 1000× SEM | 0.27 ± 0.10 | 0.25 ± 0.11 | 0.18 ± 0.06 |
| 2000× SEM | 0.17 ± 0.09 | 0.15 ± 0.05 | 0.12 ± 0.06 |
| 5500× AFM[a] | 0.50 ± 0.07 | 0.34 ± 0.03 | 0.34 ± 0.04 |

[a]Equivalent magnification for AFM, see section 2.6.

creating an artefact on the surface. The charge associated with the surface artefact causes the tip to drag the artefact along. This is shown as a white irregular structure within the AFM 2D and 3D images (figure 6). No significant difference was found between *Ra* measurements when compared to either the optical or SEM imaging method ($p > 0.05$; figures 9*a* and 10), in both the circumferential and longitudinal directions.

## 3.4. Multiscale assessment

In the circumferential orientation, when imaged by optical microscopy at low magnification (10×), $Ra_C$ was significantly greater than measurements taken at all magnifications of SEM ($p < 0.05$; figure 8).

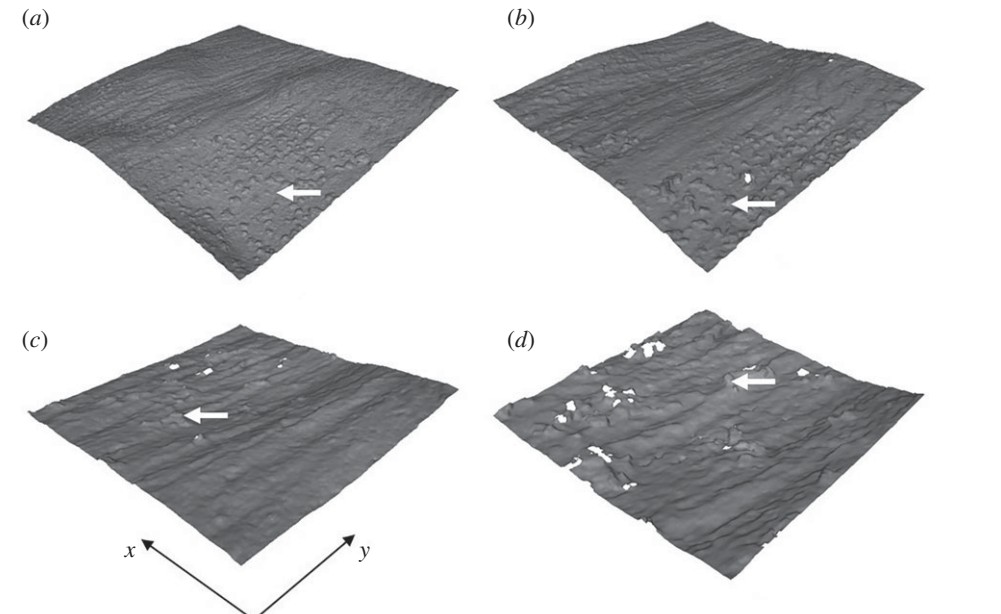

**Figure 3.** 3D reconstruction of optical images at (*a*) 10×, (*b*) 20×, (*c*) 50×, and (*d*) 100× magnification. Specimen dimensions along the *x* and *y* axes are (*a*) 1623 × 1623 µm; (*b*) 811 × 811 µm, (*c*) 323 × 323 µm, (*d*) 162 × 162 µm. Minor residue—white arrow.

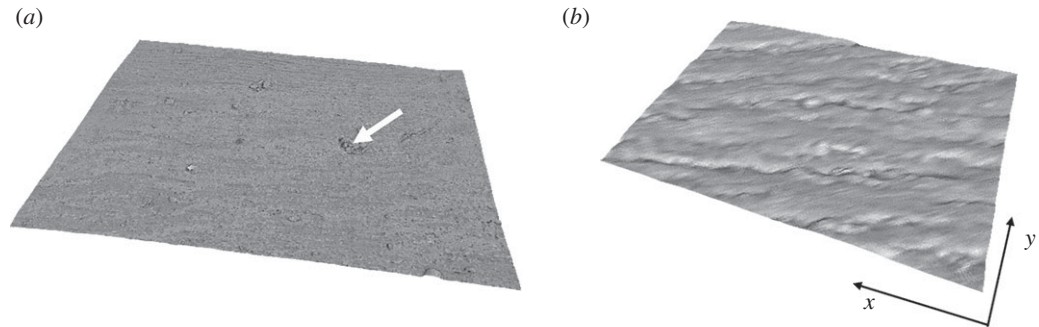

**Figure 4.** 3D reconstruction of SEM images at (*a*) 100× and (*b*) 1000× magnification. Specimen dimensions in *x* and *y* axis are (*a*) 1700 × 1230 µm and (*b*) 170 × 123 µm. Minor residue—white arrow.

However, at higher magnifications of optical microscopy (100×), no significant difference was seen in $Ra_C$ compared to when imaging at all magnifications with SEM ($p < 0.05$; figure 9*a*).

$Ra_C$ decreased with magnification, with a significant correlation to a logarithmic relationship (figure 8; equation (3.1); $R^2$ of 0.75; $p < 0.05$). When considering the optical results alone there was a significant logarithmic relationship with an $R^2$ of 0.91 (figure 9*a*; equation (3.2); $p < 0.05$), with $Ra_C$ decreasing with increase of magnification. Similarly, when assessing $Ra_{C\beta}$ from optical microscopy measurements, a logarithmic relationship was noted with $R^2$ of 0.91 (figure 9*b*; equation (3.3); $p < 0.05$). In equations (3.1)–(3.3), $\eta$ is magnification and units of $Ra$ are µm.

$$Ra_C = -0.30 \ \log_{10}(\eta) + 1.12, \tag{3.1}$$

$$Ra_{C\_OPTICAL} = -0.32 \ \log_{10}(\eta) + 1.20 \tag{3.2}$$

and

$$Ra_{C\beta} = -0.22 \ \log_{10}(\eta) + 0.82. \tag{3.3}$$

Mean and standard deviation for $Ra_L$ across all magnifications measured with all microscopy types are presented in figure 10. There was a significant difference in $Ra_L$ ($p < 0.05$; figure 10) when comparing the minimum (10×; optical) and maximum (2000×; SEM) magnifications of non-contact imaging methods (figure 10). However, there was no overall trend noted across all magnifications. $Ra_L$ had a mean of 0.26 ± 0.04 µm when removing the 10× and 2000× magnification values, which does not differ to the mean of 0.26 ± 0.06 µm calculated across all magnifications.

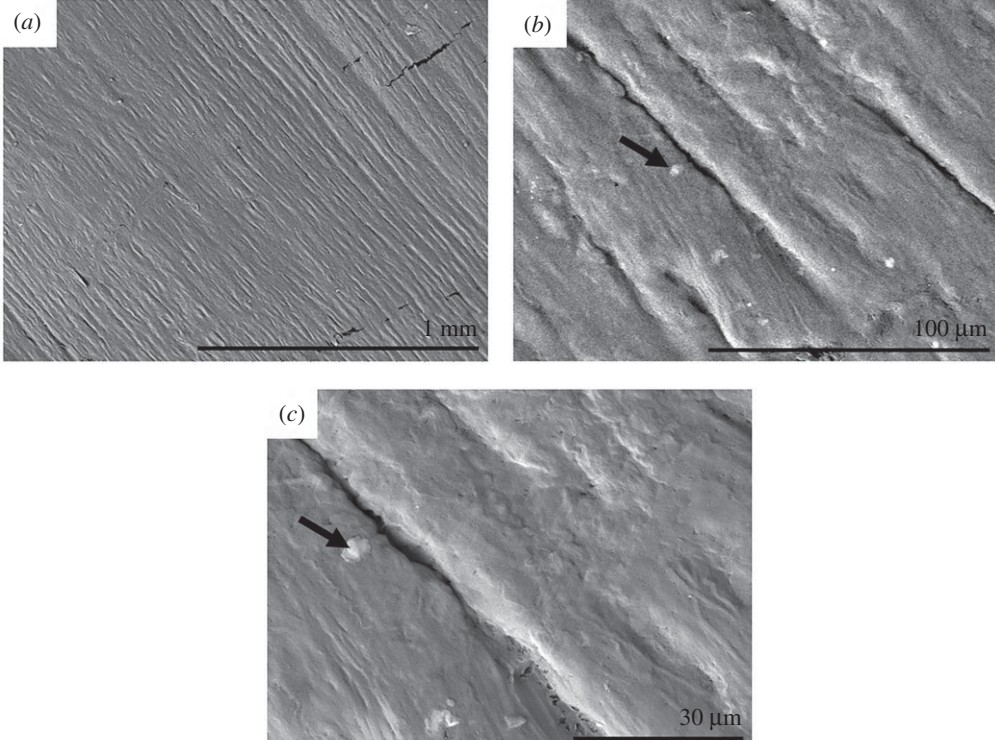

**Figure 5.** 2D SEM images at (*a*) 100×, (*b*) 1000× and (*c*) 2000× magnification. Black arrows—charging of specimen.

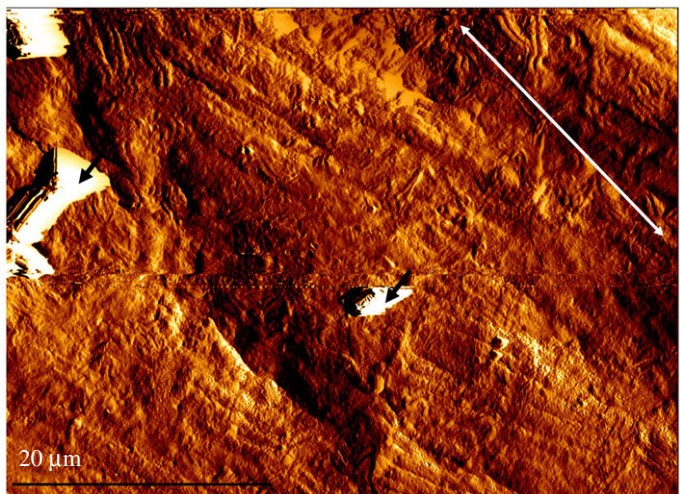

**Figure 6.** 2D AFM image. Black arrows—minor residue. White arrow—longitudinal direction.

## 4. Discussion

This study, for the first time, analyses the multiscale surface roughness of a porcine LAD coronary artery. The mean longitudinal surface roughness across light microscopy, SEM and AFM, and across magnifications ranging from 10× to 5500× was 0.26 ± 0.06 µm. However, there was a significant trend for the circumferential surface roughness, which decreased with increasing magnification from 10× to 1000×, with no further decrease to magnifications of up to 5500×. Measurements from SEM reconstruction were consistently lower than those obtained when using light microscopy or AFM. These findings highlight the importance of both microscopy type and magnification on surface roughness. However, surface roughness of LAD coronary arteries appears to be more sensitive to magnification than microscopy type.

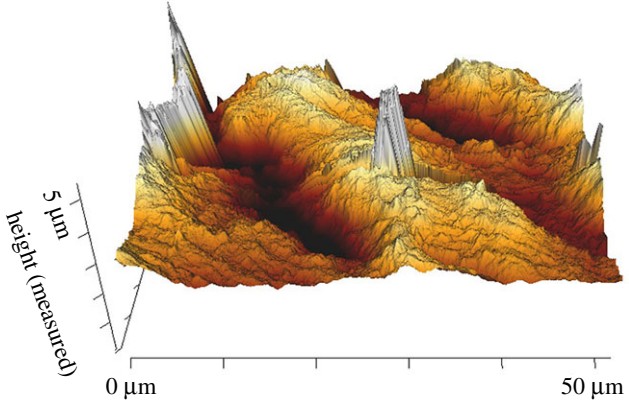

**Figure 7.** 3D reconstruction of AFM image.

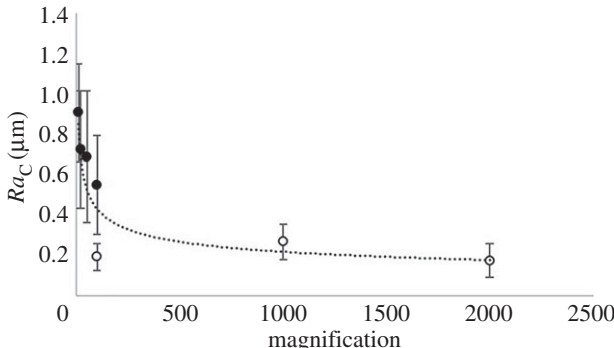

**Figure 8.** Multiscale analysis of circumferential mean surface roughness for optical (solid marker) and SEM (no-fill marker) at various magnifications (10×, 20×, 50×, 100×, 1000× and 2000×). Error bars are standard deviation. Logarithmic relationship shown.

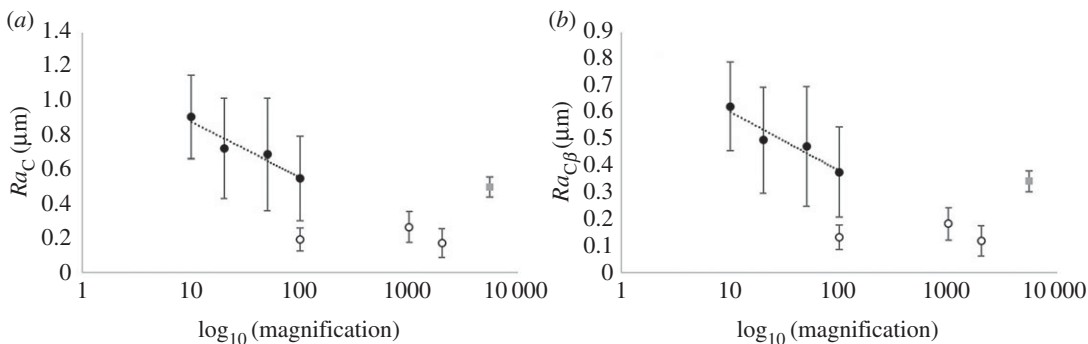

**Figure 9.** Logarithm of magnification levels to base 10 for mean data of (*a*) $Ra_C$, and (*b*) $Ra_{C\beta}$, where error bars are standard deviation (solid marker = optical, no-fill marker = SEM, grey square = AFM). Logarithmic relationship shown for optical microscopy.

Thus, a multiscale characterization of coronary arteries is essential in assessing the surface roughness from macro- and nano-scale.

The surface roughness of non-biological surfaces has been investigated, for example AFM was used to measure the effect of surface finish on $Ra$ of cardiovascular stents, while sandblasted surfaces were found to have multiscale arithmetic mean deviation ($Sa$) properties [23]. In this study for the first time the multiscale surface properties of coronary arteries was investigated. SEM consistently provided lower $Ra_C$ results than both optical and AFM, and even at like-for-like magnification (100×) a significant difference was seen between optical and SEM measurements of $Ra_C$. Consistent with our study it has been noted that measurement method can alter the surface roughness. Spencer *et al.* noted a difference in roughness when using contact and non-contact methods [24], with higher roughness measured by confocal rather than AFM. Ghosh *et al.* noted that $Ra$ of articular cartilage SEM results were higher

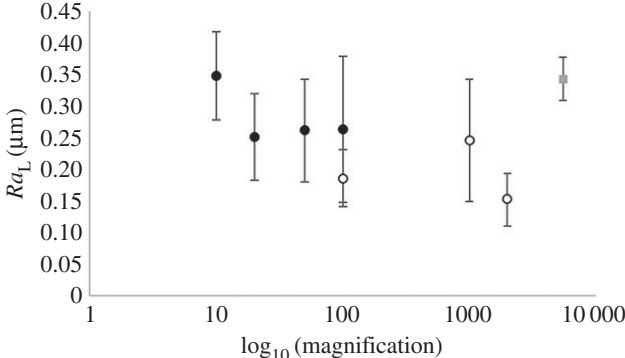

**Figure 10.** Logarithm of magnifications levels to base 10 for mean data of $Ra_L$, where error bars are standard deviation (solid marker = optical, no-fill marker = SEM, grey square = AFM). No trend noted.

than those taken with AFM [8]. However, there is no reason that the surface of articular cartilage should present the same multiscale properties as coronary arteries. The distinct multiscale roughness trends of these studies highlight the importance of multiscale assessment of surfaces. Furthermore, studies appear to agree that a consistent measurement technique should be used when making comparisons of surface roughness of biological tissue.

It is possible to identify disease in biological tissue from surface roughness [9,10,25,26]. Intimal hyperplasia results in an accumulation of smooth muscle cells beneath the endothelium, whereas vasculitis results in an inflammation of the vessel [27]. Therefore, it is hypothesized that the type and extent of coronary artery diseases could also hold a relationship to surface roughness. Investigating the surface of arteries may increase our understanding of coronary artery disease. Scale has been shown as significant when measuring properties of biological materials [10,28]. Recent studies have highlighted the capability of using multiscale biological and physiological properties of the heart for computational modelling of the circulatory system, emphasizing the potential for studying healthy and diseased cardiac systems [11]. The new generation of biological materials already use multiscale properties to influence cell growth and mechanical properties, creating micro- and nano-porosity within materials [29]. The results in this study could therefore enable bioinspired surfaces of cardiovascular devices to mimic the properties of natural arteries more closely.

Ridges were apparent along the surface of coronary arteries [3], which has implications for the helical blood flow in coronary arteries observed by other studies [30–32]. Further, atherosclerotic lesions form in a spiral pattern along coronary arteries and have been shown to affect flow resistance and wall shear stress at micro-scale levels [33]. Understanding of this helical phenomenon is relatively limited, and it is unknown whether the phenomenon reduces or increases the chance of atherosclerotic lesions to form [34]. However, the phenomenon is now considered in the design of vascular devices [12]. Physical coronary artery models have been created through additive manufacturing techniques to study blood flow in healthy and diseased systems [13]. Assessing the surface of coronary arteries at multiscale will provide invaluable information to predict the effect of multiscale roughness, for example through disease of coronary arteries, on blood flow and this helical phenomenon. Additionally, this will aid in the development of new biomaterials that can increase or decrease this phenomenon.

Multiscale measurement is important to replicate physiological function. Other studies have qualitatively assessed the dehydrated surface of coronary arteries using SEM [35,36], and they have also investigated hydrated samples using scanning force microscopy [35]. Endothelial cells were noted to be aligned longitudinally, as they were within this study, causing ridges along the sample surface [37]. However, none of these studies quantified the surface roughness of arteries, or noted a helical arrangement of ridges. As these studies did not compare the higher magnifications to lower, as in this paper, the helical arrangement may not have been seen due to the sample area assessed. For example, with AFM at 5500× it was not possible to identify this helical arrangement, only when assessing the ridges at a macro- and micro-scale.

This study demonstrates that 3D surface topology can be acquired, which can be exported as a compatible output file type for Computer Aided Design and computational modelling. Other studies have used additive manufacture to create phantom models that mimic the mechanical properties of cardiovascular tissue, to predict leakage occurrence following valve replacement [14]. Similarly, the 3D

reconstructions created in this study could be reproduced using computer aided manufacture (CAM), such as 3D printing, to create bio-mimicking coronary artery surfaces. There is also the ability to apply the surfaces to CFD models to develop more predictable fluid structure interaction models, as other studies have shown that endothelium roughness affects blood flow [38]. The reconstructions validate the feasibility of multiscale analysis of the effect of surface topography on blood flow, with a realistic representation of roughness at a relevant scale to the system [39]. Additionally, it can help to create a standard to which new biomaterials or cell seeded structures can be compared.

The dehydrated specimen surface roughness at 10× optical microscopy was lower than presented previously [3] ($Ra_C$ and $Ra_L$: $1.98 \pm 0.26$ and $1.07 \pm 0.18$ µm; $0.91 \pm 0.27$ and $0.35 \pm 0.08$ µm). However, previous work did not investigate the surface roughness in multiscale detail and the underlying trends are not affected, which are that: $Ra_C$ is significantly greater than $Ra_L$ [3,4]; dehydration increases the $Ra_C$ surface roughness [4]; an increase in magnification decreases the measured values of $Ra_C$ (figure 8).

Processing of tissue is required to enable imaging of tissue, however, this can alter the surface roughness of coronary artery [3]. The method presented previously [4] described how a correction factor was calculated for the processing of biological tissue to its dehydrated form. This could be replicated for other microscopy techniques, assessing different fixation chemicals, dehydration techniques and the effect of sputter coating on surface roughness. SEM and AFM enable investigation of the surface of coronary arteries at the nano-scale, for which a correction factor could be determined to correct the changes caused by processing to surface roughness at a nano-scale. Environmental scanning electron microscopy (ESEM) could aid measuring the surface roughness at the various stages of processing, including in a hydrated form, at higher magnifications than possible with an optical microscope [40]. Although, due to lack of conductivity with hydrated samples when using ESEM, it is not possible to study biological material at high magnification as damage is caused to the surface. Additionally, as shown in this study and elsewhere [8], using a different microscopy technique can provide varying surface roughness measurements and therefore a comparison between SEM and ESEM may not be appropriate, unless similar magnifications can be achieved. Hence, this study proposes the calculation of the uncorrected and corrected surface roughness values as outer limits. The optical Alicona system presented in this study is suitable to enable the outer limits to be calculated, as it allows surface roughness to be measured with no processing.

# 5. Limitations of the study

As discussed above, the effect of chemical processing is likely to affect the surface roughness of coronary arteries [4]. Using a correction factor in this study, it was possible to calculate the outer limits of surface roughness and the results of this study support a multiscale trend. Further, this study highlights the importance of using the same microscopy preparation protocols to allow comparison of the endothelial surface. Subsequently, assessing like-for-like magnifications using different microscopy types, with differences noted between scale of magnification.

Residual stresses, induced during excision, preparation and processing of the tissue, could result in the macro-scale 'ridges' visible in the walls of the arteries. It is unclear as to what extent these ridges affect the final trends and results, and it is noted that other studies have noted their presence too [37]. However, this potential limitation highlights an additional benefit of performing measurements over multiple scales as it is unlikely that any bias introduced by macro-scale ridges would alter measurements at the micro- and nano-scale. Further, by focusing scans on a small area of the specimen any larger scale changes in the tissue structure are reduced. Therefore, the trends obtained, and quantitative range of surface roughness reported, are expected to be representative of the roughness of porcine coronary arteries. The full extent of these limitations, though, will only be known through *in vivo* and *in situ* assessment multi-scale roughness, which is not currently feasible.

Finally, it should be noted that this study makes no comment on the effect of a bioinspired multi-scale surface on re-endothelialization [41] or neointimal hyperplasia [42,43]. Re-endothelialization is primarily driven by the abluminal attachment which falls outside the scope of this investigation. Instead, in this study we suggest that, firstly, the multi-scale surface roughness of coronary arteries should be considered for bioinspired, next generation, replacement materials (either tissue engineered or as a medical device) so as to replicate the native system. Secondly, that multi-scale surface roughness can be used as a quantitative measure for comparison for such replacement materials.

# 6. Conclusion

This is the first study to perform a multiscale analysis on the surface roughness of left anterior descending coronary arteries. Surface roughness has been shown to vary when measured at different magnification levels, with *Ra* sensitive to the microscopy method used. 3D reconstructions of the surface topology at multiscale were suitable for exporting to computer aided design software for predictive simulation or manufacture purposes. The following conclusions have been made:

— $Ra_C$ was significantly lower when calculated with SEM ($0.21 \pm 0.09\,\mu m$);
— $Ra_C$ measured by AFM ($0.50 \pm 0.07\,\mu m$) did not significantly differ to results of optical microscopy ($0.72 \pm 0.31\,\mu m$);
— $Ra_L$ did not vary with microscopy type or magnification, with an average value of $0.26 \pm 0.06\,\mu m$;
— For non-contact microscopy methods, *Ra* was significantly greater at lower ($10\times$) compared to higher ($2000\times$) magnification in both the circumferential and longitudinal direction ($0.91 \pm 0.27$ compared to $0.17 \pm 0.09$, and $0.35 \pm 0.08\,\mu m$ compared to $0.15 \pm 0.05\,\mu m$, respectively).

Ethics. Ethical approval was granted for this study by the University of Birmingham *Research Support Group* [ERN_15-0032].

Data accessibility. The datasets supporting this article have been uploaded as part of the electronic supplementary material.

Authors' contributions. H.E.B. and D.M.E. conceived the study; H.E.B. designed the study, carried out imaging investigations, collected data, performed statistical analyses and drafted the manuscript; R.C. carried out imaging investigations and contributed to the manuscript; K.J. and D.M.E. helped design the study; D.M.E. helped draft the manuscript. All authors gave final approval for publication.

Competing interests. The authors declare that they have no competing interests.

Funding. H.E.B. was funded by an *Engineering and Physical Sciences Research Council* scholarship [M114612B]. R.C. was funded by the *Defence Science and Technology Laboratory*. This study was partly funded by an *Innovation and Research Award* from the *Institute of Physics and Engineering in Medicine*. Equipment used in this study was funded by the *Science City Research Alliance (SCRA) AM2 Collaboration*.

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
