## [Reviewer comments · Royal Society Open Science]

Review History

RSOS-190915.R0 (Original submission)

Review form: Reviewer 1

Is the manuscript scientifically sound in its present form?

No

Are the interpretations and conclusions justified by the results?

No

Is the language acceptable?

Yes

Do you have any ethical concerns with this paper?

No

Have you any concerns about statistical analyses in this paper?

No

Recommendation?

Reject

Comments to the Author(s)

I am not convinced by the rebuttal, mainly regarding the two main concerns expressed earlier.

1) the surface roughness of the endothelium is not relevant for stenting. Re-endothelialization is obviously important, but the luminal surface roughness is not relevant in this context. It is the attachment of the abluminal part of the endothelial cells that determine the re-endothelialization!

2) the artifacts that are induced by processing are not only chemical by nature, there is a strong mechanical component to it. The residual stresses in the wall will cause the artery to shrink, thus inducing the well known 'ridges'. How much this contributes to the observed results will remain unknown.

Review form: Reviewer 2**Is the manuscript scientifically sound in its present form?**

Yes

Are the interpretations and conclusions justified by the results?

Yes

Is the language acceptable?

Yes

Do you have any ethical concerns with this paper?

No

Have you any concerns about statistical analyses in this paper?

No

Recommendation?

Accept as is

Comments to the Author(s)

I thank the authors for addressing my comments on the first submitted version of the manuscript.

Decision letter (RSOS-190915.R0)

26-Jun-2019

Dear Dr Burton,

The editors assigned to your paper ("Multiscale three-dimensional surface reconstruction and surface roughness of porcine left anterior descending coronary arteries") have now received

comments from reviewers. We would like you to revise your paper in accordance with the referee and Associate Editor suggestions which can be found below (not including confidential reports to the Editor). Please note this decision does not guarantee eventual acceptance.

Please submit a copy of your revised paper before 19-Jul-2019. Please note that the revision deadline will expire at 00.00am on this date. If we do not hear from you within this time then it will be assumed that the paper has been withdrawn. In exceptional circumstances, extensions may be possible if agreed with the Editorial Office in advance. We do not allow multiple rounds of revision so we urge you to make every effort to fully address all of the comments at this stage. If deemed necessary by the Editors, your manuscript will be sent back to one or more of the original reviewers for assessment. If the original reviewers are not available, we may invite new reviewers.

- Data accessibility

<http://datadryad.org/submit?journalID=RSOS&manu=RSOS-190915>

- Competing interests

- Authors' contributions

- Acknowledgements

- Funding statement

Kind regards,

on behalf of Dr Derek Abbott (Associate Editor) and R. Kerry Rowe (Subject Editor)
openscience@royalsociety.org

Associate Editor's comments (Dr Derek Abbott):

Comments to the Author:

Please add a new section to the paper called "Limitations of the Study" just before the conclusion section. In there please thoroughly discuss limitations, including but not limited to those raised by reviewers. The paper will be rejected if this is not carried out satisfactorily.

Reviewers' Comments to Author:

Reviewer: 1:

I am not convinced by the rebuttal, mainly regarding the two main concerns expressed earlier.

1) the surface roughness of the endothelium is not relevant for stenting. Re-endothelialization is obviously important, but the luminal surface roughness is not relevant in this context. It is the attachment of the abluminal part of the endothelial cells that determine the re-endothelialization!

2) the artifacts that are induced by processing are not only chemical by nature, there is a strong mechanical component to it. The residual stresses in the wall will cause the artery to shrink, thus inducing the well known 'ridges'. How much this contributes to the observed results will remain unknown.

Reviewer: 2:

I thank the authors for addressing my comments on the first submitted version of the manuscript.

Author's Response to Decision Letter for (RSOS-190915.R0)

See Appendix A.

Decision letter (RSOS-190915.R1)

07-Aug-2019

Dear Dr Burton,

I am pleased to inform you that your manuscript entitled "Multiscale three-dimensional surface reconstruction and surface roughness of porcine left anterior descending coronary arteries" is now accepted for publication in Royal Society Open Science.

on behalf of Dr Derek Abbott (Associate Editor) and R. Kerry Rowe (Subject Editor)
openscience@royalsociety.org

Appendix A

We would like to thank the reviewers and the editorial team for their time. We have addressed each point below and to aid this process have put **reviewer's comments in red** and our responses in black. Within the manuscript, changes have been highlighted in yellow, including an **address change for the corresponding author** and the addition of a **limitations section** before the conclusion.

Referee 1:

1) the surface roughness of the endothelium is not relevant for stenting. Re-endothelialization is obviously important, but the luminal surface roughness is not relevant in this context. It is the attachment of the abluminal part of the endothelial cells that determine the re-endothelialization!

The focus of this current study is not re-endothelialization following the use of a medical device. The focus is to provide a first quantitative assessment of the multiscale properties of surface roughness. The authors suggest that such measurements may have applications to enable the next generation of medical devices to be manufactured so as to be bio-inspired; this might be tissue engineered replacement constructs (which intend to fully mimic the replaced artery) or this might be a more standard stent where its surface better mimics the roughness characteristics of the native artery. There might even be future applications to assess disease (as has been shown other tissues, e.g. osteoarthritis and cartilage surface roughness). However, we do not state that improved surface roughness would alter re-endothelialization in the manuscript. To avoid any misinterpretation, by association or otherwise, we have now also limited direct reference to neointimal hyperplasia in the **abstract (lines 3-4) and on pg. 12 line 17**. Additionally, a limitations section has been included in the manuscript. The limitation in regards to the scope of the findings has been clarified, as shown below:

Finally, it should be noted that this study makes no comment on the effect of a bio-inspired multi-scale surface on re-endothelialisation [41] or neointimal hyperplasia [42-43]. Re-endothelialisation is primarily driven by the abluminal attachment which falls outside the scope of this investigation. Instead, in this study we suggest that, firstly, the multi-scale surface roughness of coronary arteries should be considered for bio-inspired, next generation, replacement materials (either tissue engineered or as a medical device) so as to replicate the native system. Secondly, that multi-scale surface roughness can be used as a quantitative measure for comparison for such replacement materials.

2) the artifacts that are induced by processing are not only chemical by nature, there is a strong mechanical component to it. The residual stresses in the wall will cause the artery to shrink, thus inducing the well known 'ridges'. How much this contributes to the observed results will remain unknown.

Tissue preparation and chemical processing may have altered the stress state of the tissue as compared to its in vivo state. It is possible that residual stresses may alter some of the measurements reported. To deal with this, firstly, the effects of chemical processing have been assessed. Returning excised ex vivo tissue to its innate stress state (which in itself is a dynamic phenomena) is a limitation of this study, and of any study where in vivo measurements cannot be made. However, the authors believe that this limitation highlights the advantage of performing multi-scale analysis because micro- and nano-scale analysis is less likely be prone to such bias than macro-scale measurements. Hence, small segments of tissue were analysed in this study, with

analysis is reported across a range of magnification. For clarification, a limitations section has been included within the manuscript, with the relevant section shown below:

As discussed above, the effect of chemical processing is likely to affect the surface roughness of coronary arteries [4]. Using a correction factor in this study, it was possible to calculate the outer limits of surface roughness and the results of this study support a multiscale trend. Further, this study highlights the importance of using the same microscopy preparation protocols to allow comparison of the endothelial surface. Subsequently, assessing like-for-like magnifications using different microscopy types, with differences noted between scale of magnification.

Residual stresses, induced during excision, preparation and processing of the tissue, could result in the macro-scale 'ridges' visible in the walls of the arteries. It is unclear as to what extent these ridges affect the final trends and results, and it is noted that other studies have noted their presence too [37]. However, this potential limitation highlights an additional benefit of performing measurements over multiple scales as it is unlikely that any bias introduced by macro-scale ridges would alter measurements at the micro- and nano-scale. Further, by focusing scans on a small area of the specimen any larger scale change in the tissue structure are reduced. Therefore, the trends obtained, and quantitative range of surface roughness reported, are expected to be representative of the roughness of porcine coronary arteries. The full extent of this limitations, though, will only be known through *in vivo* and *in situ* assessment multi-scale roughness, which is not currently feasible.